# Detection of *Toxocara cati* Larvae from Ostrich and Wild Boar Meat Intended for Human Consumption

**DOI:** 10.3390/pathogens10101290

**Published:** 2021-10-07

**Authors:** Alice Michelutti, Sofia Sgubin, Christian Falcaro, Valentina Cagnin, Alessia Zoroaster, Patrizia Danesi

**Affiliations:** Parasitology, Mycology and Medical Enthomology Laboratory, Istituto Zooprofilattico Sperimentale delle Venezie, Legnaro, 35020 Padova, Italy; ssgubin@izsvenezie.it (S.S.); cfalcaro@izsvenezie.it (C.F.); vcagnin@izsvenezie.it (V.C.); azoroaster@izsvenezie.it (A.Z.); pdanesi@izsvenezie.it (P.D.)

**Keywords:** Toxocariasis, *Toxocara cati*, meat, ostrich, wild boar, PCR

## Abstract

*Toxocara cati* is a common roundworm of cats and wild felids and, together with *T. canis*, it is the main causative agent of human toxocariasis. Humans may become infected by ingestion of embryonated eggs via contaminated soil, food, or water, or by ingestion of raw or undercooked meat of paratenic hosts that are infected by *Toxocara* larvae. In this study, we report the detection of *T. cati* larvae from meat samples of ostriches and wild boars. These samples were inspected by enzymatic digestion, as part of the trichinellosis surveillance. As ostrich meat is intended for “carpaccio” preparation, a traditional Italian raw meat preparation, there is the need to make the consumption of this meat safe. For this purpose, it is recommended to freeze the meat before preparation. Our findings confirmed that *T. cati* larvae can contaminate muscle tissues of paratenic hosts, increasing the risk of infection due to the consumption of raw or undercooked meat.

## 1. Introduction

*Toxocara cati* (Schrank, 1788) is a common ascarid roundworm of cats and wild felids. In the definitive hosts, the adult forms of *T. cati* are found in the upper tract of the small intestine and may produce thousands of unembryonated eggs, which pass in faeces, contaminating the environment [1]. Eggs are not immediately infective; embryonation occurs over weeks to months under optimal conditions of temperature and humidity, and embryonated eggs can remain infective in the environment over months to years [2]. The widespread contamination of the environment with *Toxocara* eggs facilitates infection of both definitive and paratenic hosts, including humans.

Embryonated eggs represent the infective form for the definitive host and for a wide range of paratenic hosts, such as rodents, lagomorphs, ruminants, pigs, and birds. After ingestion by paratenic hosts, larvae penetrate the intestinal wall and migrate via blood system to several organs, including liver, lungs, muscles, and the Central Nervous System (CNS) [3,4,5,6].

Human toxocariasis is a parasitic zoonosis, mainly caused by *T. canis* and *T. cati* [5].

People can be infected by *T. canis* and *T. cati* through accidental ingestion of embryonated eggs via contaminated soil, food, or water, or by eating raw or undercooked meat containing *Toxocara* larvae [7].

Although the larval stages are unable to develop to adult worms in humans, infective *Toxocara* larvae may migrate to a range of tissues, causing damage to whatever tissue they happen to enter, resulting in a number of clinical manifestations such as visceral larva migrans (VLMs), ocular larva migrans (OLMs), covert or common toxocariasis (CT), and neurotoxocariasis (NT) [8,9,10,11,12,13]. With any migration paths taken by *Toxocara* larvae, however, a majority of clinical manifestations caused by *Toxocara* infection is asymptomatic or non-specific; therefore, its impact on public health could be underestimated [9,14].

However, the importance of this parasitic zoonosis is increasing in the last few years, and the Centers for Disease Control and Prevention (CDC) listed human toxocariasis among the five most neglected parasitic diseases worldwide [14,15,16].

*Toxocara* larvae can be occasionally detected in non-definitive hosts by enzymatic digestion of muscle, collected during routine meat inspection at the slaughterhouse and sent to the laboratory for official controls for *Trichinella* [17]. Indeed, according to the European legislation “carcasses of horses, wild boar and other farmed and wild animal species susceptible to *Trichinella* infestation shall be systematically sampled in slaughterhouses or game-handling establishments as part of the post-mortem examination” (EU Regulation 1375/2015) [18].

In Italy, due to increasing popularity of “meat carpaccio”, (consisting of thinly sliced raw meat) the presence of *Toxocara* larvae might represent a source of infection for humans. In this study, we report the detection and molecular identification of *T. cati* larvae in meat samples of ostrich and wild boars, which were inspected by enzymatic digestion, for trichinellosis control.

## 2. Results

*Trichinella* larvae were not detected in ostriches or wild boar meat. However, nematode larvae (n = 8), belonging to genus other than *Trichinella*, were detected in meat samples of two ostriches in September 2019 and in one ostrich in December 2020. All ostriches had been reared in the same farm, sited in the province of Venice (northeastern Italy).

Nematode larvae (n = 4) were also found in meat samples of wild boars in December 2020. All larvae were alive and motile when observed under the stereomicroscope. Larvae measured 400–450 µm in length and 15 µm in width. The anterior part of the body ended in a sub-terminal mouth, while the posterior one tapered to a slender tail. The esophageal region, occupying one-third of the total length, and the more opaque intestinal region were clearly distinguishable. Based on the morphological characters, all larvae were classified as other than *Trichinella*, which have a greater length (800–900 µm) and a different anatomic structure [19].

Molecular identification confirmed the species as *Toxocara cati* (Figure 1). Phylogenetically, sequences of the Internal Transcribed Spacer 1 (ITS1) region were invariably identical among ostrich and wild boar, with a single sequence type (ST) observed. In a tree (Figure 2), sequences of *Toxocara cati* clustered into highly supported (bootstrap value = 100%) clades, clearly separated from other *Toxocara* species.

## 3. Discussion

In this study, we report the detection of *T. cati* larvae in ostrich and wild boar muscles. Although *T. cati* larvae have been already detected in birds and domestic pigs, including chickens and wild avian species of the genera *Corvus*, *Falco*, *Circus*, and *Buteo* [6,17], the detection of *T. cati* larvae in ostrich and wild boar muscle poses a higher risk of infection for humans, as this meat can be consumed “raw”. 

Several cases of human toxocariasis occurred after the consumption of raw meat and organs of paratenic hosts, especially raw liver of different animals, including lamb [20] and ostrich [21]. In particular, the consumption of raw ostrich liver caused NT in a young boy, who showed symptoms of eosinophilic meningitis and lungs and liver involvement [21].

In all these cases, the diagnosis of human toxocariasis was based on clinical and serological findings, concluding *T. canis* was the etiological agent. However, immunological tests show cross reactivity between *T. canis* and *T. cati*, probably resulting in an over-diagnosis of *T. canis*-associated infection [2]. This hypothesis might also be supported by results of this study in which all *Toxocara* larvae belonged to *T. cati* species and not to *T. canis.*

Animals included in our study were raised at an extensive farm. These animals might have become infected by ingesting embryonated eggs that contaminated in the pasture, or by ingesting other paratenic hosts, such as invertebrates and small rodents. 

*Toxocara* larvae were detected in ostrich meat samples in September 2019 and September 2020. Ostriches were still positive for the presence of *T. cati* larvae in muscle, one year after the first detection. This could be due to the contamination of the pasture with *T. cati* eggs, which may remain infective in the environment for years [2], or to a recontamination of the pasture with feces from stray cats.

The growing habit of eating raw or undercooked meat, typical of many cultures [21,22], significantly increases consumer exposure to zoonotic parasites which can cause severe disease [23]. Our findings emphasize that the consumption of raw meat increases the risk of *Toxocara* spp. infection. For this purpose, it is recommended to store the meat at −20 °C for at least 48 h, considering that this treatment proved to be effective for inactivating *T. cati* larvae from chicken meat [24]. A similar preventive measure is used for fish species at risk of infection by nematodes larvae of the family Anisakidae, according to the European regulation (CE Regulation 853/2004) [25].

## 4. Materials and Methods

### 4.1. Enzymatic Digestion

From September 2019 to December 2020, 24,098 muscle samples were controlled for *Trichinella* at the laboratory of Parasitology, Mycology, and Medical Entomology (IZSVe - Legnaro, Italy). These samples included 15,804 pigs, 8291 horses and wild boars, and 3 ostriches, and were all collected by the veterinary authority at slaughterhouse as part of the post-mortem examination according to the European Regulation [18]. 

The analysis consisted of different steps: (i) enzymatic digestion of pooled meat samples (100 g for pool), followed by (ii) filtration, and (iii) sedimentation and observation of the digested material under the stereomicroscope for the detection of nematode larvae according to the EC Regulation 1375/2015 [18] and ISO 18743:2015 [26].

Nematode larvae were individually transferred in 2 mL tube (one larva per tube) using a pipette with a 100 µL tip and stored at minus 20 °C until molecular analysis was performed.

### 4.2. Molecular Tools

DNA was extracted from a single larva using a commercial kit (DNeasy Blood & Tissue Kit; Qiagen, Hilden, Germany), following the manufacturer’s recommendations. A negative control, consisting of PBS (phosphate buffer solution), was included systematically in each series of DNA extractions. DNA was amplified by PCR, targeting a portion of the Internal Transcribed Spacer 1 region (ITS1) gene. 

The following ITS1 primers were used: 18s-ITS1-5.8S for (5′-TAACAAGGTTTCCGTAGGTG-3′)–18S-ITS1-5.8S rev (5′-AGCTRGCTGCGTTCTTCATCGA-3′) amplifying 650–750 bp [27]. 

Reactions were performed in 50 μL of mix containing AmpliTaq Gold™ DNA Polymerase with Buffer II and MgCl_2_, 0.5 μM of primers, 0.2 μM of dNTP (Thermo Scientific), and 5 μL of extracted DNA. The PCR cycle is as follows: an initial denaturation step at 95 °C for 10 min, then 35 amplification cycles (denaturation at 94 °C for 30 s, annealing at 55° for 30 s, extension at 72 °C for 30 s) and a final elongation step at 72 °C for 10 min.

All positive amplicons were sequenced, as described previously [28], for taxonomic confirmation. Molecular phylogeny was performed using Neighbor joining method (Jukes Cantor model) on the ITS1 (MZ598516 – MZ597544) sequence data sets MEGA v.6.0. Bootstrap values shown at the main nodes represent the probabilities based on 1000 replicates. We added sequences of *Toxocara cati*, *T. canis*, *T. vitulorum*, and *T. tanuki* available from GenBank (https://www.ncbi.nlm.nih.gov/nucleotide, accessed on 4 August 2021), using *Ascaris lumbricoides* as an outgroup.

## Figures and Tables

**Figure 1 pathogens-10-01290-f001:**
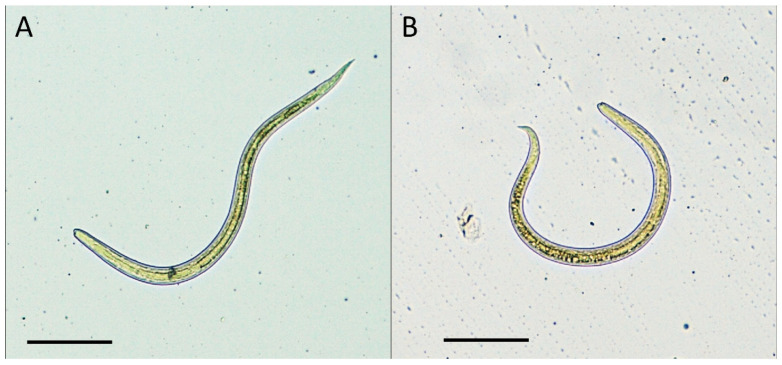
*Toxocara cati* larvae detected in muscle samples of ostriches (**A**) and wild boars (**B**) by artificial enzymatic digestion. Scale bars 100 µm.

**Figure 2 pathogens-10-01290-f002:**
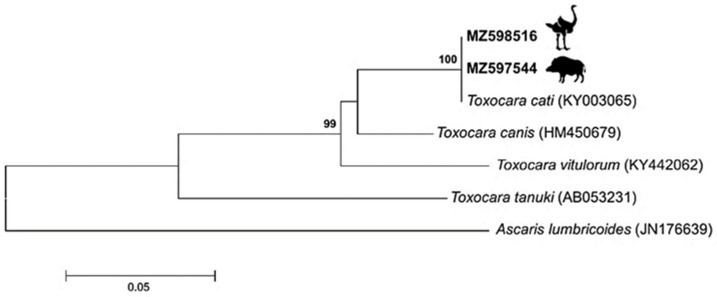
Phylogenetic tree based on Internal Transcribed Spacer 1 (ITS1) sequences of *Toxocara* species. The tree was constructed using the Neighbor joining method (Jukes Cantor model). Bootstrap values above 90%, indicated at the main nodes, represent the probabilities based on 1000 replicates. The sequences obtained from our study are shown with accession number in bold. *Ascaris lumbricoides* sequence was used as an outgroup.

## Data Availability

Not applicable.

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
