# Peer review of "Detection of Toxocara cati Larvae from Ostrich and Wild Boar Meat Intended for Human Consumption"

_pathogens, 2021, doi:10.3390/pathogens10101290_

Round 1

Reviewer 1 Report

This manuscript demonstrated a larval stage of Toxocara cati in ostrich and wild boar meats through enzymatic digestion technique, which was an interesting finding. However, this manuscript was somewhat disorganized, and it was difficult to understand why this finding is significant or worthy of publication. To be able to publish this manuscript, I believe authors need to have better descriptions and explanations on several major issues listed below.

  • Toxocara larvae have been already detected in “pigs” and “birds” as this current manuscript even mentioned in the line 31. So then why is it so important to find Toxocara cati larvae in ostrich (birds) and wild boars (pigs)? Please explain more details why your findings are significant. 
  • Sometimes some facts/information were mixture of T. cati and T canis. If applicable, I recommend focusing superficially and exclusively only on Toxocara cati to make this manuscript stronger. If not feasible, then I recommend clearly stating/separating information from T. canis and T. cati
  • I am not familiar with carpaccio or the enzymatic digestion test on striated muscled regulated by the European legislation, and many other readers may not be able to understand the situation in Europe, either. I strongly suggest including (1) how common carpaccio is (I have never heard of it) or how often raw/undercooked meats get consumed in Europe – how risky is it; (2) how important to freeze meats before preparation even though the enzymatic digestion test has been applied as meat inspection in Europe

(Minor revision suggestions)

Lines 12-14: I think this sentence, “In this study, we report …. by enzymatic digestion.” was incomplete, and also somewhat confusing. Please rewrite this sentence.

Lines 35-36: I did not find the fact “human toxocariasis is a parasitic zoonosis listed among the five most neglected parasitic diseases” from your reference #7 (CDC website) – please double check if this reference was accurate.

Lines 37-38: “..it is considered to be the most common human parasitic infection in the United States, particularly among the impoverished.” – I did find this sentence from your reference #9; however, this same reference also mentioned as “Recent research indicates that toxocariasis is the most common human parasitic worm infection in the United States, affecting millions of American living in poverty.” (second paragraph in “1.1 Global Burden of Disease”).  To me, it is a big difference between “the most common parasitic infection” vs. “the most common parasitic worm infection.” Please be more specific and provide us an accurate statement. Also, the sentence in lines 37-38 in this current manuscript is almost exact the same to the sentence I found in this reference #9. Please rewrite your sentence.  

Line 40: The reference #2 for this sentence, “…the evidence for the zoonotic potential of T. cati was recently documented.” was actually published in 2003. I am not sure if we call this reference as “recently” (it's been almost 2 decades ago!). I understand what you were trying to say (yes, zoonotic potential of T. cati has been underestimated), but I would recommend rewording this sentence.   

Lines 51-52: I am not sure if your reference #17 can fully support for your statement of “In Europe, toxocariasis resulted to be the most prevalent zoonotic nematodiasis,” because this retrospective survey was done only 1523 cases in 2000-2016. Probably you can reword this statement and be more specific and accurate. Also, this data was not specific to T. cati (included both T. cati and T. canis). Are there any references that demonstrated only T. cati? If yes, I would recommend citing those specific data.   

Lines 61-64: Please add a reference for the sentence of “Among non-zoonotic nematodes, ……. are included.”

Line 69: I recommend mentioning as “Toxocara cati larvae” instead of just "Toxocara larvae" here because there are publications demonstrated Toxocara canis larvae in wild boars.

Line 72: Somehow “Material and Methods” was moved to the very end of this manuscript. Please change the order of this manuscript.

Line 79: Please include all specific morphologies of Toxocara spp. – how did you morphologically differentiate Toxocara larvae from other differentials? Also please include references.

Line 121: You stated, “In chickens, T. cati larvae behaved much differently than those of T. canis,” however, there were no statements explaining the differences between T. cati and T. canis behaviors in chickens.

Line 144: Please be specific on a number of the muscle samples instead of saying “more than 24,000” as this is a scientific manuscript (I think it should be 24,098 samples).

Author Response

Reviewer 1

This manuscript demonstrated a larval stage of Toxocara cati in ostrich and wild boar meats through enzymatic digestion technique, which was an interesting finding. However, this manuscript was somewhat disorganized, and it was difficult to understand why this finding is significant or worthy of publication. To be able to publish this manuscript, I believe authors need to have better descriptions and explanations on several major issues listed below.

  • Toxocara larvae have been already detected in “pigs” and “birds” as this current manuscript even mentioned in the line 31. So then why is it so important to find Toxocara cati larvae in ostrich (birds) and wild boars (pigs)? Please explain more details why your findings are significant.

Although T. cati larvae have been already detected in birds, including chickens and wild avian species of the genera Corvus, Falco, Circus, and Buteo, the detection of T. cati larvae in ostrich muscle poses a higher risk of infection for humans, since ostrich meat was intended to be eaten raw, as a traditional raw preparation called “carpaccio”. On the other hand, chicken meat is usually consumed cooked in our country, therefore the detection of T. cati larvae in chickens does not represents a risk factor for human toxocariasis. The detection of T. cati larvae in wild boars meat represents a risk of infection for humans too, since wild boar meat, as well as domestic pig meat, is often consumed raw, as cured products (now lines 111-119).

  • Sometimes some facts/information were mixture of cati and T canis. If applicable, I recommend focusing superficially and exclusively only on Toxocara cati to make this manuscript stronger. If not feasible, then I recommend clearly stating/separating information from T. canis and T. cati.

Thank you for suggestion, but it was not feasible to focus on T. cati only, especially when I speak about human toxocariasis. In most cases, human toxocariasis was due to T. canis, but it seem to be over-diagnosed, because of the cross-reactivity of the immunological test between T. canis and T. cati. To make it clearer, I specified when I was referring to T. cati only or both T. canis and T. cati (highlighted in lines 35-40).

  • I am not familiar with carpaccio or the enzymatic digestion test on striated muscled regulated by the European legislation, and many other readers may not be able to understand the situation in Europe, either. I strongly suggest including
  • (1) how common carpaccio is (I have never heard of it) or how often raw/undercooked meats get consumed in Europe – how risky is it;
  • (2) how important to freeze meats before preparation even though the enzymatic digestion test has been applied as meat inspection in Europe.
    • Carpaccio is a traditional Italian dish consisting of thinly sliced meat or fish, served raw, but there are several traditional raw preparations in European countries (tartare in France, mett in Germany, pastrami in Romania, etc…) (now lines 146-147).
    • Freezing can be considered as a sanitizing measure for meat intended to be eaten raw or undercooked, when infected with zoonotic nematodes (now lines 159-160).

(Minor revision suggestions)

Lines 12-14: I think this sentence, “In this study, we report …. by enzymatic digestion.” was incomplete, and also somewhat confusing. Please rewrite this sentence.

Lines 12-14 rephrased as suggeted. In this study, we report the first detection of T. cati larvae from meat samples of ostriches and wild boars. These samples were analyzed by enzymatic digestion, as part of the trichinellosis surveillance.

Lines 35-36: I did not find the fact “human toxocariasis is a parasitic zoonosis listed among the five most neglected parasitic diseases” from your reference #7 (CDC website) – please double check if this reference was accurate.

Reference checked as suggested. https://www.cdc.gov/parasites/npi/ (accessed on 24 August 2021).

Lines 37-38: “it is considered to be the most common human parasitic infection in the United States, particularly among the impoverished.” – I did find this sentence from your reference #9; however, this same reference also mentioned as “Recent research indicates that toxocariasis is the most common human parasitic worm infection in the United States, affecting millions of American living in poverty.” (second paragraph in “1.1 Global Burden of Disease”).  To me, it is a big difference between “the most common parasitic infection” vs. “the most common parasitic worm infection.” Please be more specific and provide us an accurate statement. Also, the sentence in lines 37-38 in this current manuscript is almost exact the same to the sentence I found in this reference #9. Please rewrite your sentence.

Lines 37-38 rephrased. However, the importance of this parasitic zoonosis is increasing in the last years and it was listed among the five most neglected parasitic diseases, according to the Centers for Disease Control and Prevention (CDC). Although the infection is worldwide distributed, the highest seroprevalence are described in developing countries. Humans in rural areas seem most likely to be infected than semirural and urban areas [17] (now lines 50-55).

Line 40: The reference #2 for this sentence, “…the evidence for the zoonotic potential of T. cati was recently documented.” was actually published in 2003. I am not sure if we call this reference as “recently” (it's been almost 2 decades ago!). I understand what you were trying to say (yes, zoonotic potential of T. cati has been underestimated), but I would recommend rewording this sentence.

Line 40 rephrased as suggested. Although, there has been an assumption that T. canis was the most likely cause of human toxocariasis, the zoonotic potential of T. cati could have been underestimated, probably because immunological tests show cross-reactivity between T. canis and T. cati [2] (now lines 35-40).

Lines 51-52: I am not sure if your reference #17 can fully support for your statement of “In Europe, toxocariasis resulted to be the most prevalent zoonotic nematodiasis,” because this retrospective survey was done only 1523 cases in 2000-2016. Probably you can reword this statement and be more specific and accurate. Also, this data was not specific to T. cati (included both T. cati and T. canis). Are there any references that demonstrated only T. cati? If yes, I would recommend citing those specific data.

Unfortunately, there are no data specific exclusively for T. cati. The sentence was rephrased as following. “In Europe, a systematic review of the literature published from 2000 to 2016, showed that toxocariasis was the most prevalent zoonotic nematodiasis. Although most cases were due to T. canis, the prevalence of this species may be over-estimated, as the diagnosis was confirmed by immunological tests, which can result in a cross-reaction between the T. canis and T. cati [18] (now lines 55-60).

Lines 61-64: Please add a reference for the sentence of “Among non-zoonotic nematodes, ……. are included.”

Reference added (line 72).

Line 69: I recommend mentioning as “Toxocara cati larvae” instead of just "Toxocara larvae" here because there are publications demonstrated Toxocara canis larvae in wild boars.

Amended as suggested.

Line 72: Somehow “Material and Methods” was moved to the very end of this manuscript. Please change the order of this manuscript.

This order is required by the journal.

Line 79: Please include all specific morphologies of Toxocara spp. – how did you morphologically differentiate Toxocara larvae from other differentials? Also please include references.

The sentence was rephrased as following: … based on the morphological characters, all larvae were classified as other than Trichinella, especially because of the larvae lenth. Indeed, Toxocara larvae are smaller than Trichinella larvae (450-600µ vs 800-900µ respectively) [21]. Other larvae morphological details were no evident after digestion of specimens (now lines 87-89).

Line 121: You stated, “In chickens,T. cati larvae behaved much differently than those of T. canis,” however, there were no statements explaining the differences between T. cati and T. canis behaviors in chickens.

The sentence has been deleted.

Line 144: Please be specific on a number of the muscle samples instead of saying “more than 24,000” as this is a scientific manuscript (I think it should be 24,098 samples).

Revised with 24.098 (now line 166).

Reviewer 2 Report

In their manuscript, Michelutti et al. reported for the first time Toxocara cati larvae in ostrich and wild boar meat, supported by DNA sequencing data. If these hosts were already suspected to be paratenic hosts for Toxocara spp., based on serology and case reports, this article brings a scientific proof of their possible presence as living larvae in the meat. From my point of view, this simple report is highly informative concerning the ways of contamination with Toxocara spp. and moreover with T. cati. Additionnally, the whole manuscript is well written and I have only few suggestions of modifications:

L69: "underlying"

L70: "raw"

L83-88: For many nematode genera, interspecific variations in ITS and 18S rRNA genes are insufficient for species-level identification. Mitochondrial DNA, and classically the cox1 gene, is often used for a more accurate identification. I would suggest to perform additionnal DNA sequencing of this target, in order to compare the obtained sequences to those from Genbank (e.g. see Fava NMN, Cury MC, Santos HA, Takeuchi-Storm N, Strube C, Zhu XQ, Taira K, Odoevskaya I, Panovag O, Mateus TL, Nejsum P. Phylogenetic relationships among Toxocara spp. and Toxascaris sp. from different regions of the world. Vet Parasitol. 2020 Jun;282:109133. doi: 10.1016/j.vetpar.2020.109133.)

L89-97 (and L177-183): How many bootstrap replicates and which substitution model?

L137-140: This conclusion seems to me highly pertinent: when regarding the European regulation (EC N° 853/2004) fish products intended to be eaten raw should be frozen at least 24h before dish preparation, but not mammal meat (nor poultry meat, but which is unfrequently consumed raw or undercooked)...

L151-154: I have a practical question concerning this part: how do you transfer the larvae in 2 mL tube? Is, for a given sample, each larva isolated from other ones, or are they pooled together? In other terms, my question is: do you extract DNA from a unique larva?

Author Response

Reviewer 2

In their manuscript, Michelutti et al. reported for the first timeToxocara cati larvae in ostrich and wild boar meat, supported by DNA sequencing data. If these hosts were already suspected to be paratenic hosts for Toxocara spp., based on serology and case reports, this article brings a scientific proof of their possible presence as living larvae in the meat. From my point of view, this simple report is highly informative concerning the ways of contamination with Toxocara spp. and moreover with T. cati. Additionnally, the whole manuscript is well written and I have only few suggestions of modifications.

Thank you very much with your kind words.

L69: "underlying

L70: "raw"

Amended as suggested.

L83-88: For many nematode genera, interspecific variations in ITS and 18S rRNA genes are insufficient for species-level identification. Mitochondrial DNA, and classically the cox1 gene, is often used for a more accurate identification. I would suggest to perform additionnal DNA sequencing of this target, in order to compare the obtained sequences to those from Genbank (e.g. see Fava NMN, Cury MC, Santos HA, Takeuchi-Storm N, Strube C, Zhu XQ, Taira K, Odoevskaya I, Panovag O, Mateus TL, Nejsum P. Phylogenetic relationships among Toxocara spp. and Toxascaris sp. from different regions of the world. Vet Parasitol. 2020 Jun;282:109133. doi: 10.1016/j.vetpar.2020.109133.)

Thank you for the helpful suggestion. Indeed, in our laboratory we use to confirm the “contaminat” nematode larvae by molecular protocol (by Marucci et al., 2013) using only ITS1/2 and 18S primers. We thank reviewer 2 for the kind suggestion and in future, we will implement our molecular protocol including the cox1 gene PCR as suggested. At the moment, we are sorry we are not able to implement the manuscript with additional DNA sequencing of cox1 gene.

89-97 (and L177-183): How many bootstrap replicates and which substitution model?

Amended as suggested. Molecular phylogeny was performed using Neighbor joining method included in MEGA v.6.0. Bootstrap values above 90% are indicated above the branches in Figure 2, Figure 3 and in figure captions.

L137-140: This conclusion seems to me highly pertinent: when regarding the European regulation (EC N° 853/2004) fish products intended to be eaten raw should be frozen at least 24h before dish preparation, but not mammal meat (nor poultry meat, but which is unfrequently consumed raw or undercooked)...

According to Reviewer 1 suggestions, the sentence has been slightly modified (now lines 159-160).

L151-154: I have a practical question concerning this part: how do you transfer the larvae in 2 mL tube? Is, for a given sample, each larva isolated from other ones, or are they pooled together? In other terms, my question is: do you extract DNA from a unique larva?

Larvae were individually transferred in 2 ml tube (one larva per tube) by 100 µl pipette (now lines 175-176). We extracted and amplified DNA from single larva (now line 179).

Reviewer 3 Report

FIle attached

Author Response

Reviewer 2

General comments

The authors show the detection and molecular identification of larvae found in ostrich and wild boar

meats using enzymatic digestion technique within trichinellosis surveillance program.

The topic is interesting because it provides additional data on the presence of T. cati larvae in ostrich and wild boar meat. However, the ms setting and the description of the research appear a bit “pretentious” and, on its whole unclear as it is difficult to understand where the novelty of such contribution is and the background related to the research.

Authors thank the reviewer for suggestions. In the manuscript authors described for the first timeToxocara cati larvae in ostrich and wild boar meat, supported by DNA sequencing data. Although these hosts were already described to be paratenic hosts for Toxocara spp., this MS brings a scientific proof of their possible presence as living larvae in the meat. The aim of authors was to writing a simple report informative concerning the ways of contamination with Toxocara spp. and moreover with T. cati., with suggestion that meat intended to be eaten raw should be frozen at least 48h before dish preparation. Althought ostrices meat are unfrequently consumed raw or undercooked internationally, we have evidence that ostriches meat preparations (including ostriches carpaccio) can be part of the restaurant menu, at least in Italy. Trying to make the reading of this MS easier we shortened the text of 20% and we submitted the MS as Brief report as suggested by the reviewer.

Thus, the contribution does not entirely convince the reader for the following reasons:

Title

The title seems unclear. Did the authors know what the destination of that meat exactly was when they processed the meat? If so, it is undefined in the text, too. Then, why do they omit the wild boars?

We have updated the title in the MS including wild boar as suggested. “Detection of Toxocara cati larvae from ostrich and wild boar meat intended for human consumption”.

The laboratory of Parasitology, Micology and Medical Entomology at the IZSVe perfoms official controls for Trichinella (by enzymatic digestion) of muscle collected from carcasses during routine meat inspection at the slaughterhouse, including ostriches meat samples. All samples are collected from animal species “susceptible to Trichinella infestation” as part of the post-mortem examination according to the European Regulation (EU Regulation 1375/2015). All those specimens are from carcasses of animals intended for human consumption, including raw preparation as “carpaccio” which is popular in this part of Italy.

Introduction

Line 61-68:

-“to prevent human trichinellosis”: the authors have to specify why the IZS practices include ostrich

surveillance for Trichinella control.

-“from domestic animals”: this statement is incorrect as not all the “domestic animals” are investigated for the presence of trichinella.

At the laboratory we process all meat specimens collected at the slaughterhouse that veterinarian authorities send to check for Trichinella, both following the EU legislation for swine, equids and wild boars and for all other farmed animals for which the veterinarian authorities have considered a possible health risk.

The following sentence was now in the Introduction section “According to the European legislation “carcasses of horses, wild boar and other farmed and wild animal species susceptible to Trichinella infestation shall be systematically sampled in slaughterhouses or game-handling establishments as part of the post-mortem examination” (EU Regulation 1375/2015)” (lines 51-55).

 -“trichinellosis surveillance”: to my best knowledge, ostrich are NOT included in the list of animals

to be checked.

The sentence has been corrected, specifying “farmed animal susceptible to Trichinella” (lines 52-55).

If IZS is called to check also ostriches, why only three samples in one year do they have checked?

The IZS laboratories have been designated by the Ministry of Health as official laboratories for Trichinella diagnosis. However, our role is “passive” and all the actions related to sampling, data registration, sending to the lab and release of the meat is demanded to the local health authority.

The consumption of ostrich raw meat is likely a novelty in this area and therefore the veterinarian of the slaughterhouse asked for trichinella examination.

Line 77: “meat samples of ostrich”: Since ostrich are not undertaken to control for the presence of

trichinella, the authors have to clarify why this species has been included in the IZS control practices.

We discussed about this in previously responses.

 Is the IZS called to check ostrich meat only when the meat is for sure served for "carpaccio" preparation? It sounds pretty strange to me. We know that ostrich is used to make fillets, hamburgers, steaks, salami, hams, roast beef, stews, ragouts etc. etc.

No, IZS is called to check the meat that official veterinarians send to lab, i.e swine, equids, wild boars (intended for human consumption) and all other farmed animals that can be susceptible to Trichinella. In this last case, the local health authority organized the sampling. Other animals are checked in the frame of surveillance programs, such as foxes, mustelids and birds of prey.

Line 80: “other than Trichinella”: if the study has been carried out in the framework of “Trichinella

control program” (which is not plausible) what about Trichinella results?

This is now updated in results section “Trichinella larvae were not detected in ostriches or wild boar meat” (line 62).

Line 91 (Figure 1): this should be placed not here but above, where non-Trichinella larvae are described. It should be better placed below (line 95), as only after molecular analysis you identified the species (see Figure 1 legend).

Amended as suggested.

Line 107: Again, I think that such detection has been done NOT “within a Trichinella control program”. The authors should report the reason why they decided to check the ostrich meat.

Toxocara larvae have been detected after enzymatic digestion of meat samples. Specimens were sampled by veterinary authority at the slaughterhouse as part of the post-mortem examination, according to the European Regulation 1375/2015 and sent to the laboratory for Trichinella control. M&M section is now updated as following “From September 2019 to December 2020, 24098 muscle samples were controlled for Trichinella at the laboratory of Parasitology, Mycology and Medical Entomology (IZSVe - Legnaro, Italy). These samples included 15,804 pigs, 8,291 horses and wild boars and 3 ostriches and were all collected by the veterinary authority in slaughterhouses as part of the post-mortem examination according to the European Regulation [18] (lines 123-127).

Lines- 110-111: Again, I do not think ostrich's meat is only intended for carpaccio preparation. If this was the reason why the detection was needed, the authors have to specify this aspect.

We discussed about this in previous responses.

Lines 118-119: This sentence can be removed. It is unnecessary.

We thank for suggestion but we prefer to keep this sentence in the text because describes a case of meningitis in a young men caused by Toxocara canis after ingestion of raw ostrich liver. This is relevant because asserts the importance of ostrich as source of Toxocara infection.

Lines 124-125: Such conclusion is unclear if the readers basically cannot understand why the authors have checked the safety of meat ostriches.

We have already discussed about this in previously responses .

Line 135: Repetition

Lines 147-149: Repetition

Lines 150-151: Redundant/unnecessary sentence

The discussion section has been shortened and cleared of repetitions.

M&M

Line 157: Again, it is unclear why ostriches have been undertaken to trichinella control. If so, why

so low number of ostriches are checked in a year?

See the above comments

CONCLUSION

The reviewer would suggest the Authors to reorganize the ms, trying to be more cautious and more

explicit about the framework in which such research is placed and the real novelty of such detection,

in ostriches and wild boar. The ms needs to be streamlined before it can be considered for publication. In my opinion, after its reorganization and removal of useless, not straightforward and redundant sentences the contribution can be resubmitted as a Short note.

Trying to make the reading of this MS easier we shortened the text of 20% and we submitted the MS as Brief report as suggested by the reviewer.

Round 2

Reviewer 1 Report

Some improvements are observed; however, I recommend further revisions to continue improving this manuscript.

Abstract:

Line 14-16: The sentence “Since ostrich meat was intended ….., there was the need to make the consumption of this meat safe.” is confusing.

I thought ostrich meat is still consumed raw, correct? If yes, I recommend changing the tense of this sentence, “Since ostrich meat is intended …., there is the need to make the consumption of this meat safe.”

Line 15: I believe the location of comma is wrong here. It should be “carpaccio,” (instead of “carpaccio”,). Please correct here and others (I saw the same mistake in several different sentences) throughout this manuscript.

Line16-17: I recommend switching to “Our findings confirmed that T. cati larvae can contaminate …”

Introduction

Line 25-27: I would recommend rewriting this long sentence. Maybe something like "Eggs are not immediately infective; embryonations occurs over weeks to months under optimal conditions of temperature and humidity, and embryonated eggs can remind infective in the environment over months to years." (This is just suggestion. Please leave it or take it)

Line 30: Recommend removing “Indeed”

Line 40: Recommend removing “Significantly”

Line 50-52: Which country (or countries) considers toxocariasis as one of the five neglected parasitic diseases? In the USA? Or worldwide? Please be specific.

Line 56-59: Please add a reference (or references) for the sentence, “Although most cases were … between T. canis and T. cati.”

Line 75-79: This entire paragraph is a conclusion. This paragraph should not be in “Introduction” section. Please remove this paragraph and include your study objectives here.

Also, just for your information – the location of period was wrong here too. It should be “carpaccio.” (instead of “carpaccio”.)

Results

Line 85-89: I still request including a brief description (a sentence or two) of those nematode larvae (size, shape of head, tail, and any other morphological characteristics) here or for Figure 1.  

Discussion

Line 114: It should be “…since ostrich meat is intended to” (instead of “was”). Also, it should be “carpaccio.” (instead of “carpaccio”.)  

Line 115-117: I am very questionable about this sentence “On the other hand, … for human toxocariasis.” I am not sure if this sentence was authors’ personal opinion or real scientific fact based on data. I would probably reword this sentence, and a take-home message for readers will be something like “there are some risks to consume any type of raw meats – ostrich, chickens, wild boars, and any other game meats,” instead of ostrich and wild boar meats are more dangerous than chickens. Or simply remove this sentence. If you would like to keep this sentence as it is, please include a reference (or references) to support what it describes.  

Line 117-119: Can nematode larvae survive within the cured meats? Is there any reference that demonstrated viable nematode larvae or other parasites within the cured products?

Line 132: It should be “…raised at an extensive farm.”

Line 132-133: I would recommend rewording this sentence as “These animals had the possibility to ingest embryonated eggs from the contaminated pasture or to ingest infected vertebrate and invertebrate paratenic hosts, …”

Line 135: I am not sure about this sentence “This, heavy environment contamination was the origin of infection.” How do you know the environment was contaminated by Toxocara cati eggs? Did you test the pasture? Were there many cats in/around the pasture? Please be clearer.  

Line 136: Please remove “Indeed”

Line 139: Please correct as “carpaccio,

Line 139: Please correct as “there is the need…”

Line 140: Please correct as “it is recommended to store…”

Line 146-147: I am not sure if this sentence, “Moreover, pets are … including Toxocara spp.,” can support your hypothesis why detection of toxocariasis is getting more common in adults, instead of children. If pets are dewormed regularly, then it is going to be less contamination and lower prevalence of toxocariasis in humans in general. Please correct here as needed.

Line 147-148: I am not sure if I could agree with this sentence, “the growing habit of eating raw or undercooked meat typical of many cultures.” Yes, I agree with you – there are still some traditions/cultures that people consume raw meats although now we know consumption of raw meats can cause some parasitic diseases. However, I am not sure if the tradition/habit of raw meat consumption has been “growing.” If this is truly happening, please use proper reference(s).

Probably you can discuss whether or not “increasing detection of toxocariasis in adults could have been due to better, more advanced diagnostic techniques.”     

Line 150-154: I recommend re-writing this long sentence.

First, “it may be recommended” (line 151) or actually you do recommend freezing meats? Earlier in this manuscript, you kept saying “it is recommended,” however, now you say, “it may be recommended.” It is confusing. Please be clear.

Would you recommend consuming the meats that contained zoonotic parasites? This sentence “…it may be recommended to freeze the meat, which is positive for zoonotic nematodes…” is confusing. If the meat was positive for zoonotic nematodes, I would recommend discarding the meat, or definitely the meat should be confirmed parasite-free by freezing, heating, cooking, etc. before consumption.

Need a reference (or references) for the sentence “…adopted for fish….nematodes of the family Anisakidae.”

I also recommend including a very short paragraph to discuss and re-emphasize it is risky to consume raw meats and is recommended freezing the meats because the enzymatic digestion test cannot detect all zoonotic parasitic infections because the technique is applied only in a limited sample (small meat samples) and game meats (by hunters) do not need to go through the enzymatic digestion test.

Materials and Methods

Line 165-169: I recommend re-write this long (almost too long) sentence.

Author Response

I thank the reviewer for corrections and suggestions.
Please see the attachment.

Reviewer 2 Report

I thank the authors for their corrections, however, one of my concerns was not adressed:

89-97 (and L177-183): How many bootstrap replicates and which substitution model?

When publishing a phylogenetic analysis, authors shall detail the parameters they input (number of bootstrap resampling, model of substitution). Please provide them. 

Also, by checking more carefully some of the references, I saw that the sequences AM412316 and AM411108 from the figure 3 are in fact sequences of the mitochondrial DNA of T. malaysiensis and T. canis, respectively. As 18S rRNA is genomic, the new sequences obtained from the larvae in this study  (MZ598517 and MZ596339) cannot align with them. And, as I suspected in my first comment, these new 18S rRNA sequences do not allow species identification of the larvae (BLAST research : 99% identity with T. cati but also with many other Ascaridia, even out of the Toxocara genus).

Actually, this means that the figure 3 is false (EF180078 is 99% identical to MZ598517, thus should be close to it), so all parts dealing with 18S rRNA should be deleted. Also, an explanation for this tree is welcome.

Author Response

I thank the reviewer for corrections and suggestions.

Reviewer 3 Report

The Authors have clarified the reviewers' concerns.

I am glad the Authors have now submitted the ms as a Brief report which is more acceptable given the data provided. Well done.

Just a final piece of advice:

Lines 54-55: with reference to EU Regulation 54 1375/2015, I didn’t find such a sentence in it but in the following: COMMISSION REGULATION (EC) No 2075/2005 of 5 December 2005.

If the Authors confirm, please, replace the reference.

Author Response

The Authors have clarified the reviewers' concerns.

I am glad the Authors have now submitted the ms as a Brief report which is more acceptable given the data provided. Well done.

Just a final piece of advice:

Lines 54-55: with reference to EU Regulation 54 1375/2015, I didn’t find such a sentence in it but in the following: COMMISSION REGULATION (EC) No 2075/2005 of 5 December 2005.

If the Authors confirm, please, replace the reference.

We thank the reviewer for the kind words and for the suggestions that helped us to improve the MS.

The reference 18 “European Commission. Commission Implementing Regulation (EU) 2015/1375 of 10 August 2015 laying down specific rules on official controls for Trichinella in meat. OJ. 2015, 7–34” contains the sentence reported in text “carcasses of horses, wild boar and other farmed and wild animal species susceptible to Trichinella infestation shall be systematically sampled in slaughterhouses or game-handling establishments as part of the post-mortem examination” in the chapter I, article 2.2.

The same sentence is also reported in the previous Commission Regulation 2075/2005 in the chapter I, article 2.3 that was abrogated and replaced by the EU Regulation 1375/2015.

We prefer to keep the reference 18 as it is in the MS, because the Commission Regulation (EC) No 2075/2005 of 5 December 2005 is no longer in force, with date of end of validity: 30/08/2015 (https://eur-lex.europa.eu/eli/reg/2005/2075/oj).

Round 3

Reviewer 1 Report

I see the improvement on this manuscript. Thank you for working on this manuscript very hard. I have only a few, very minor suggestions/recommendations.

(Abstract)

Line 12: I would recommend modifying as “…of paratenic hosts that are infected by Toxocara larvae.”

Line 13: I would say “inspected by” fits better here instead of “analyzed by”

(Introduction)

Line 23: I recommend saying “T. cati are found” instead of “T. cati live.”

Line 40: I think “T. canis and T. cati” fits better here, instead of “T. canis or T. cati.”

Line 47-48: I recommend re-writing this sentence “Anyway, the disease …. could be underestimated.” I still do not think the word “anyway” fits well here (very vague). I included an example sentence below.

“With any migration paths taken by Toxocara larvae, however, a majority of clinical manifestations caused by Toxocara infection is asymptomatic or non-specific; and therefore, its impact on public health could be underestimated.”

Line 49: Could you please be slightly more specific for “in the last years” here? Is it “last several years?,” “last decade years?,” etc.

Line 53: I recommend re-writing this sentence as “Humans in rural areas seem more likely infected than ones in rural and urban areas."    

Line 57: I am not sure if the word “confirmed” fits well here because the meaning of “confirm” is “establish the truth or correctness” and I know you tried to say immunological tests could have been imprecise due to cross-reactivity between T. canis and T. cati. I would say “… as the diagnosis was made by…” instead.

Line 77: I would recommend using “inspected” instead of “analyzed” here.

(Discussion)

Line 124: I would recommend using “supported” instead of “confirmed” here. “Confirmed” is such a strong word, and I am not sure if you can strongly state your hypothesis was actually confirmed by your study.

Line 127: It should be “These animals might have become….”

Line 128: Please correct as “…that contaminated in the pasture,”

Line 140: Please include a comma, “…considered a childhood disease, Kwon et al. observed…”

Line 141: Please correct as “The authors considered this shift was due to life style changes.”

Line 142: I would suggest correcting this sentence as “…children often play indoor rather than outdoor,” maybe you can use a different word, such as “mostly” and “commonly,” instead of “often” here too.

Author Response

We thank the reviewer for the suggestions. Specific comments are reported in the attached file.
